# Landscape of Epidermal Growth Factor Receptor Heterodimers in Brain Metastases

**DOI:** 10.3390/cancers14030533

**Published:** 2022-01-21

**Authors:** Malcolm Lim, Tam H. Nguyen, Colleen Niland, Lynne E. Reid, Parmjit S. Jat, Jodi M. Saunus, Sunil R. Lakhani

**Affiliations:** 1Faculty of Medicine, UQ Centre for Clinical Research, The University of Queensland, Herston, QLD 4029, Australia; m.lim@uq.edu.au (M.L.); c.niland@uq.edu.au (C.N.); Lynne@crunkhorn.com (L.E.R.); 2Flow Cytometry and Imaging Facility, QIMR Berghofer Medical Research Institute, Brisbane, QLD 4006, Australia; tamhong.nguyen@qimrberghofer.edu.au; 3Department of Neurodegenerative Disease and MRC Prion Unit, UCL Institute of Neurology, Queen Square, London WC1N 3BG, UK; p.jat@prion.ucl.ac.uk; 4Pathology Queensland, Royal Brisbane Women’s Hospital, Herston, QLD 4029, Australia

**Keywords:** brain metastases, HER family, HER dimers, breast cancer, proximity ligation assay

## Abstract

**Simple Summary:**

HER2+ breast cancer patients are treated with agents that tag HER2+ tumour cells for elimination by the immune system, down-modulate HER2 activity and/or block the formation of HER2 dimers, including the neuregulin-1 receptor, HER2-HER3. HER2-targeted therapies prolong survival by lowering the risk of relapse, but do not prevent brain metastases. The reasons for this are not fully understood. We quantified HER2-HER3 dimers in 203 brain metastases, and 34 primary breast tumour samples. Dimer frequency was relatively high in brain metastases from breast, ovarian, lung and kidney cancers, and in brain metastases compared to patient-matched breast tumours; but did not reliably correlate with HER2/HER3 expression or activation. In in vitro experiments, pertuzumab failed to suppress HER2-HER3 dimers in HER2+ breast cancer cells provided with a saturating concentration of neuregulin-1. These findings may provide insights about the differences in intracranial versus extracranial efficacy of HER2-targeted therapies.

**Abstract:**

HER2+ breast cancer patients have an elevated risk of developing brain metastases (BM), despite adjuvant HER2-targeted therapy. The mechanisms underpinning this reduced intracranial efficacy are unclear. We optimised the in situ proximity ligation assay (PLA) for detection of the high-affinity neuregulin-1 receptor, HER2-HER3 (a key target of pertuzumab), in archival tissue samples and developed a pipeline for high throughput extraction of PLA data from fluorescent microscope image files. Applying this to a large BM sample cohort (*n* = 159) showed that BM from breast, ovarian, lung and kidney cancers have higher HER2-HER3 levels than other primary tumour types (melanoma, colorectal and prostate cancers). HER2 status, and tumour cell membrane expression of pHER2(Y^1221/1222^) and pHER3(Y^1222^) were positively, but not exclusively, associated with HER2-HER3 frequency. In an independent cohort (*n* = 78), BM had significantly higher HER2-HER3 levels than matching primary tumours (*p* = 0.0002). For patients who had two craniotomy procedures, HER2-HER3 dimer levels were lower in the consecutive lesion (*n* = 7; *p* = 0.006). We also investigated the effects of trastuzumab and pertuzumab on five different heterodimers in vitro: HER2-EGFR, HER2-HER4, HER2-HER3, HER3-HER4, HER3-EGFR. Treatment significantly altered the absolute frequencies of individual complexes in SKBr3 and/or MDA-MB-361 cells, but in the presence of neuregulin-1, the overall distribution was not markedly altered, with HER2-HER3 and HER2-HER4 remaining predominant. Together, these findings suggest that markers of HER2 and HER3 expression are not always indicative of dimerization, and that pertuzumab may be less effective at reducing HER2-HER3 dimerization in the context of excess neuregulin.

## 1. Introduction

Metastatic breast cancer (MBC) remains a significant social and economic burden in many countries around the world. In Australia, it is the second most common cause of cancer-related death in women [1]. Metastasis to the brain is a particularly serious complication because it produces challenging neurological side effects and is difficult to control with existing local and systemic therapies. Around 14% of all MBC patients experience symptomatic brain recurrence requiring treatment [2]. The cost of care for this group is more than double that of MBC patients without brain involvement, and they have significantly worse clinical outcomes [3,4].

Brain metastasis is common for breast tumours with amplification of the HER2 gene, *ERBB2* [2,5,6]. Not only do HER2+ patients have an elevated risk of already having brain metastases (BM) when initially diagnosed with breast cancer [7], they are also more likely to die as a direct consequence of this complication [8]. HER2 is one of four human epidermal growth factor receptor (HER) tyrosine kinases (RTKs). Dimerization amongst these receptors is activated by epidermal and neuregulin growth factor ligands (EGF/NRG), promoting pro-oncogenic signalling via the PI3K and MAPK pathways in a variety of malignancies, including BM [9,10]. Several mechanisms are thought to contribute to BM pathogenesis in HER2+MBC, including an intrinsic propensity for growth in the brain, which acts as a sanctuary site for early outgrowth [11]. The natural history of HER2+BM also has an iatrogenic component. Although HER2-targeted therapy can delay intracranial progression, paradoxically, it is also associated with the brain being the first site of recurrence, and a higher incidence of BM overall [12,13,14,15,16]. Lapatinib and trastuzumab are used in the metastatic setting because they prolong survival, but this is primarily by stabilizing extracranial disease [12,17,18,19]. 

Unfortunately, none of the existing HER2-targeted agents (lapatinib, neratinib, pertuzumab, trastuzumab, or ado-trastuzumab emtansine) can prevent brain recurrence [20]. The reasons for this appear to be complex, including patchy drug uptake due to unique pharmacokinetics in the brain compartment [21,22,23,24,25], as well as brain-specific acquired resistance. For example, brain-derived neuregulins (NRG) provide a stimulus for the HER3-PI3K cell survival axis [26,27,28]. This is supported by studies demonstrating increased expression and activation of NRG receptors, HER3 and HER4, in BM compared to matching primary breast and lung tumours [29,30,31,32]. NRG1 has a very strong binding affinity for HER3-HER2 heterodimers [33], and HER3 is an exceptionally strong activator of the PI3K cell survival axis, with six docking sites for PI3K’s regulatory substrate p85 [34,35]. 

HER3 gene (*ERBB3*) amplification and overexpression are associated with resistance to a variety of cancer drugs, including chemotherapy, hormone therapy and trastuzumab [27,36,37,38,39,40,41,42,43,44]. With increasing recognition of its importance in therapeutic resistance, a variety of HER3-targeted agents are under development [45]. HER3′s intracellular domain has low kinase activity and the conformational change required to engage p85 is dependent on transphosphorylation by other RTKs; predominantly, HER2. Hence, most agents typically target its ligand-binding domain or interfere with hetero-dimerization. There is accumulating data rationalising the testing of these agents in the brain-metastatic setting, however the relationships between receptor expression, activation and dimerization in human BM tumours have not been established. 

In this study, we optimised a proximity ligation assay pipeline for detection and analysis of HER2-HER3 complexes in human BM tissue samples. We also tested the effects of trastuzumab and pertuzumab on the HER2/HER3 heterodimer landscape of HER2+ breast cancer cell lines provided with a saturating concentration of neuregulin-1. Our findings provide insight about the reduced intracranial efficacy of HER2-targeted agents.

## 2. Materials and Methods

### 2.1. Cell Line Selection and Cell Culture

Gene copy-number (Absolute), RNAseq (log2 TPM+1) and reverse phase protein array (RPPA) expression data for HER2 and HER3 were downloaded from depmap portal (Cancer Cell Line Encyclopaedia, Broad Institute [46,47]), and used to select four breast cancer cell lines for this study (Appendix A). *ERBB2* copy number alterations in these lines are all associated with complex chromosomal rearrangements [48]. MDA-MB-361, SKBr3, MCF7 and MDA-MB-231 were sourced from the American Type Culture Collection, maintained in a supplemented medium according to recommendations, and routinely checked for mycoplasma using the MycoAlert kit (Lonza), as described previously [49].

Cells were routinely cultured on glass coverslips to 70–80% confluence, followed by overnight culture in low-serum media (1% foetal calf serum). For neuregulin assays, cells were cultured in low-serum medium supplemented with human NRG1 (R&D systems; 50 ng/mL; ‘activation medium’) for 30 min. For antibody blocking experiments, serum-deprived cells were pre-incubated (4 h) with activation medium containing trastuzumab and/or pertuzumab (Roche, 20 µg/mL; donated by the Royal Brisbane and Women’s Hospital Pharmacy). Treated cells were rinsed with phosphate-buffered saline (PBS), fixed with paraformaldehyde (4% in PBS) and permeabilised (0.5% Triton X-100, 5 min).

### 2.2. Clinical Samples and Human Research Ethics Committee Approval

Two FFPE tissue microarray (TMA) cohorts were used in this study: Mixed BM cohort: various histological origin: 48 from lung cancer, 33 breast, 30 melanoma, 19 renal cell, 17 colorectal, 5 prostate, 4 ovarian, 1 adenoid cystic, 1 neuroendocrine and 1 thyroid cancer case. HER2 status for these tumours was determined previously using immunohistochemistry (IHC) and diagnostic scoring criteria [30].Breast-BM cohort: 41 brain-metastatic breast cancer cases (34 primary breast tumours and 44 BM, of which 28 are ‘matched pairs’) [31]. HER2 status was extracted from primary breast cancer pathology reports, determined by in situ hybridization (ISH).

TMAs were constructed by sampling tumour-rich regions from each tissue block in duplicate, as cores of 0.6–1.0 mm diameter. TMAs were sectioned (4 µm) and dewaxed prior to heat-induced antigen retrieval (Table 1). After optimisation (Appendix A), PLA was performed as described for fixed cell lines, then sections were imaged with a Vectra 3 automated quantitative pathology imaging system (Perkin Elmer) at 20× magnification (emission/excitation: 461/359 nm (DAPI) and 618/590 nm (PLA)). Raw spectral image files (.im3) were spectrally unmixed to isolate individual channels and remove background autofluorescence prior to downstream image analysis. 

All samples in this study pre-date the use of pertuzumab for metastatic breast cancer. This study was approved by human research ethics committees of the National Hospital for Neurology and Neurosurgery (London, UK; mixed BM TMA construction), the Royal Brisbane and Women’s Hospital and The University of Queensland (Queensland, Australia; matched TMA construction and molecular analysis of both cohorts). 

### 2.3. Immunofluorescence

Antibodies were selected based on manufacturer guarantee and/or published evidence of specificity (Table 1). We then optimised detection conditions using immunofluorescence. For fixed cells on coverslips, non-specific binding sites were blocked (10% goat serum in PBS) for 30 min, then samples were incubated with primary antibodies in blocking buffer (Table 1) for 1 h at room temperature. After rinsing with PBS, detection was performed with Alexa Fluor™ 488-conjugated, species-specific secondary antibodies (# A28175, #A11012, Thermofisher, Waltham, MA, USA) diluted 1:400 in blocking buffer (30 min, room temperature). Samples were washed with PBS, counterstained with DAPI and mounted (Prolong Gold, #MAN0009669, Thermofisher, Waltham, MA, USA). For formalin-fixed, paraffin-embedded (FFPE) samples, sections were subjected to heat-induced antigen retrieval prior to immunofluorescent staining (Table 1).

### 2.4. Fluorescent In Situ Proximity Ligation Assay (PLA)

PLA was carried out in humidified slide chambers, according to the technical protocol and kit reagents supplied (Duolink In Situ Red Starter Kit Mouse/Rabbit, #DUO92101, Sigma Aldrich, St. Louis, MO, USA). Briefly, non-specific antibody binding sites were blocked for 1 h at 37 °C, then samples were incubated with primary antibodies raised in different host species (Table 1). For detection of dimer complexes, oligonucleotide-conjugated, species-specific secondary antibodies (supplied in the Duolink kit) were applied (1 h, room temperature), followed by ligase and ligation oligonucleotides (30 min, 37 °C) to catalyse the circularization of complementary probes bound within approximately 40 nm. Fluorescent labelled oligonucleotides and polymerase were then added (100 min, 37 °C), for “rolling-circle” amplification of dimer signals, generating a quantifiable fluorescent red spot [50]. After amplification, samples were counterstained with DAPI, and coverslips were mounted on glass slides (Prolong Gold, #MAN0009669, Thermofisher, USA). Fifteen randomly selected microscopic fields from each sample were imaged at 20x magnification (Zeiss Axio Imager M1, Carl Zeiss, Jena, Germany), generating a .czi file for each region-of-interest (ROI).

### 2.5. Digital Image Analysis

We developed a pipeline to automate the extraction of PLA data from fluorescent images. It uses two freely available bioimaging analysis software packages: QuPath (v 0.1.2), which has algorithms for nuclear identification, cell counting and batch analysis [51]; and ImageJ; specifically the “Find Maxima” plugin, which we found to be superior to other alternatives for differentiating true fluorescent foci (signals) from noise. With slightly different steps for fixed cells and tissue samples, this pipeline is depicted in Figure 1, and a detailed protocol with all the required scripts are provided as Appendix A.

Image files were loaded into QuPath (we used .czi files, though Qupath will accept other formats). An optimized cell detection algorithm was then applied on the blue channel (DAPI) to detect cells based on nuclear size and fluorescence intensity. Files were then exported to ImageJ where the optimized Find Maxima algorithm was applied to detect dimer signals in the red channel in the form of Alexa Fluor™ 594-positive foci. For FFPE tissue sections, tumour-rich ROIs were first annotated manually with the aid of matching hematoxylin and eosin-stained, serial sections, with cell and PLA signal detection restricted to these ROIs.

After verifying the accuracy of this process on test images, we developed a high throughput batch-processing approach to parse hundreds of files (Appendix A). PLA data can be expressed as signals /cell, and/or signals per unit area. We used Prism 8 (GraphPad, v9.2) for statistical analyses (specific tests and p-values indicated in the respective figure legends).

## 3. Results

### 3.1. Validation of Fluorescent In Situ PLA and Development of a Data Extraction Pipeline

We optimised a PLA protocol using breast cancer cell lines with differing HER2 and HER3 activation status, and relevance to human BM [46,47] (Figure 1a and Appendix A):SKBr3, a model of genomically driven HER2-HER3-PI3K pathway activation with *ERBB2* and *ERBB3* copy-number gains, high expression, and strong activation under baseline culture conditions, a HER3 tyrosine kinase domain mutation of unknown significance, and homozygous deletion of *PTEN.* PTEN exerts negative feedback on PI3K and is frequently lost in breast cancer BM [30,52,53].MDA(-MB-)361, a breast cancer BM-derived line with *ERBB2* and *ERBB3* gene copy-number gains but average to low baseline expression and activation. This line has dual *PIK3CA* mutations (E545K, K567R): helical domain alterations that decouple PI3K’s catalytic and regulatory domains, disrupting negative feedback [54,55,56]. Its growth in the brain is more dependent on PI3K and HER3 than HER2 [27].MCF7, a model of luminal-like breast cancer with below average baseline expression and activation of HER2 and HER3, *ERBB2* copy number loss and *PIK3CA^E545K^*.

After optimising detection conditions for the chosen anti-HER antibodies (Table 1 and Appendix A), we performed PLA analysis on these cell lines before and after treatment with neuregulin-1 (NRG1) to verify the expected frequencies of HER2-HER3 complexes. As a typical experiment involves acquiring and counting foci from at least 10 regions-of-interest (ROIs) per sample, we developed a pipeline to streamline and automate the extraction of PLA data from image files (Figure 1b). This experiment showed that HER2-HER3 complexes were detectable in low-serum conditions, with negligible differences between SKBr3, MDA361 and MCF7 cells. However, 30 min after NRG1 treatment, there were marked differences consistent with the genomic and expression status of HER2 and HER3 in these lines (Figure 1c). These data indicated that with further optimisation, this pipeline should be suitable for analysis of human clinical samples.

### 3.2. HER2-HER3 Complexes Are Prevalent in Brain Metastases with Different Histological Origins, but Do Not Correlate Reliably with Protein Level or Phosphorylation

HER2 and HER3 are induced and activated in BM, but the frequency of dimer complexes in relation to expression and activation has not been assessed. This is relevant because HER2-HER3 dimerization is therapeutically targetable with pertuzumab, but there have been no clinical studies directly testing its intracranial efficacy. We adapted the PLA pipeline for FFPE tissue sections, first optimising detection conditions (Figure 2a), then adding a tissue segmentation step that restricts cell and PLA signal counting to tumour-rich ROIs (Figure 2b). We quantified tumour-specific PLA signals per unit area and found a very strong correlation with signals per cell (in cross-section), hence either method would convey the same information for highly cellular ROIs (Figure 2c). 

Next, we applied the PLA pipeline to a human tumour sample cohort comprising BM from breast, lung, kidney, colorectal, ovarian, skin (melanoma) and prostate cancers. PLA signals were clearly detectable across the sample cohort, albeit at lower frequency than in vitro experiments as analysis was based on 4 µM tissue sections versus whole cells. 

There were 159 assessable cases, which we referenced against histologically normal tissue samples that were processed in parallel (lymph node, kidney, and colon, which express HER3 but not HER2 [57]). Data were extracted from an average of 3989 cells per sample (range 382–13,704), corresponding to an average ROI area of 350 μm^2^ (range 30–830 μm^2^). BMs from melanoma, kidney, lung, ovarian and breast cancers had significantly more frequent HER2-HER3 dimers than normal reference tissues (*p* ≤ 0.01 by Kruskal–Wallis test; Figure 2d). Breast cancer BMs had the highest frequency, though the levels in ovarian, lung and kidney cancer BM were only marginally lower (*p* > 0.01 by Kruskal–Wallis test). Levels were lowest in the prostate cancer BMs, consistent with other reports that HER receptor expression is relatively low in prostate cancers [58,59].

Separating breast and lung cancers according to HER2 status highlighted that on average, HER2+ BM had significantly more HER2-HER3 dimers than negative or equivocal cases (Figure 2e). A total of 48.5% (16/33) and 14.6% (7/48) of breast and lung BMs in this cohort were HER2+. However, the HER2-HER3 dimer frequency in HER2+ tumours was not consistently higher (Figure 2d), suggesting that HER2 IHC is not a reliable indicator of HER2-HER3 dimerization. How this relates to the efficacy of pertuzumab is currently unknown.

Finally, we determined whether HER2-HER3 dimer frequency is reflected by the extent of HER2 and HER3 phosphorylation in the breast and lung cancer BMs, by correlating PLA data with pHER2 and pHER3 IHC data that we published previously from this cohort [30]. There were indeed positive associations, particularly for breast cancer BM, but these trends were largely driven by the HER2+ cases (Figure 2f). This suggests that like HER2 status, the extent of HER2 and HER3 activation evident by IHC is also not a reliable indicator of HER2-HER3 dimer frequency in BM. 

### 3.3. HER2-HER3 Dimers Are Induced in BM Compared to Matched Primary Breast Tumours

HER2 and HER3 are induced and activated in BM, however changes in HER2-HER3 dimer frequency after metastasis to the brain has not been directly assessed. We investigated this using PLA analysis of a unique cohort of breast cancer BM with patient-matched primary breast tumours, arrayed in TMA format [31]. There were 76 assessable tumour samples from 41 patients, including 28 matched pairs. 

Overall, HER2-HER3 signal frequency was significantly higher in BMs compared to breast tumours (Mann–Whitney *p* = 2.1 × 10^−4^; Figure 3a). The difference in HER2-HER3 signals between matched primary breast and BM cases were calculated as fold-change and compared. Separating the matched pairs into subtypes defined by ER, PR and HER2 status showed average increases of 17.5, 5.9 and 2.3-fold for HER2+, triple-negative and ER+ cases, respectively (Figure 3b). In terms of case numbers, this represents increases of more than 2-fold for 11/14 (79%) HER2+ cases, 4/9 (44%) triple-negative cases and 3/4 (75%) ER+ cases. There was no statistical difference between the groups. For the HER2+ cases, 6 samples (4 patients) were from patients treated with trastuzumab prior to BM resection, and 8 were from patients who were naïve to HER2-targeted therapy at the time of BM resection. There was a marginal difference between these groups, with HER2-HER3 complexes more frequent overall in pre-treated tumours (*p* = 0.1; Figure 3c). 

For the 7 patients who had more than one BM resected, we were also able to quantify changes between successive brain tumours. Interestingly, dimer frequency was lower for second recurrences (*p* = 0.006; Figure 3d). The average interval between BM1 and BM2 in this cohort was just under 8 months (interval range 0.2–17.7 months). Two patients in this group had their BM1 and BM2 craniotomy procedures less than a month apart, hence their BM1 and 2 samples were more likely to be from synchronous than sequential recurrences, but even after excluding these patients (*n* = 5, interval 7–17.7 months), the decrease in HER2-HER3 dimer frequency between BM1 and BM2 was still apparent (*p* = 0.02).

For the two HER2+ patients in this group, there were substantial increases in dimer frequency between the second and third events, regardless of prior treatment with HER2-targeted therapy: patient-1 (very high HER2-HER3 in BM3) was heavily pre-treated, while patient-2 (high HER2-HER3 in BM3) was not treated with trastuzumab and lapatinib until after neurosurgical resection (Figure 3d). 

### 3.4. Effects of HER2-Targeted mAbs on the HER2/3 Heterodimer Landscape

Next, we investigated the effects of trastuzumab (Tz) and pertuzumab (Pz) on HER2 and HER3 heterodimer configurations, under controlled experimental conditions. We used our in vitro PLA pipeline to quantify five different complexes in SKBr3 and MDA361 cells—both *ERBB2*-amplified but with different pathway alterations and drivers. In HER2-3 driven SKBr3 cells, NRG1 induced substantial changes in the dimer landscape, marked by increases in HER2 dimerization with NRG1 receptors HER3 and HER4 (~7-fold), at the expense of HER3-EGFR complexes (reduced by ~32-fold; Figure 4a). A similar response was seen in PI3K-dependent MDA361 cells, though at a lower magnitude (Figure 4a).

Next, we incubated the cells with Tz and/or Pz prior to NRG1 stimulation, to quantify changes in the dimer landscape in the context of neuregulin saturation. Interestingly, in SKBr3 cells this did not reduce the frequency of any of the complexes, instead *increasing* all dimers except HER2-HER3 (Figure 4b). This was observed for mAb-treated cells even in the absence of NRG1 (Appendix A). However, changes in the individual complexes were inconsequential to the overall distribution (Figure 4c). For instance, Pz and Pz+Tz induced significant increases in HER3-HER4 dimerization, but this complex was very infrequent in SKBr3 cells compared to HER2-HER3 and HER2-HER4. In summary, NRG1-stimulation significantly increased the levels and relative proportions of HER2-HER3 and HER2-HER4 complexes, and this distribution was relatively unperturbed by Tz and Pz. 

MDA361 cells are *ERBB2*-amplified but not fully dependent on HER2, owing mutated PI3K that is constitutively active and decoupled from RTK feedback. Pre-treatment of this line with Tz and/or Pz produced modest but statistically significant changes in HER2/3 dimer frequency, notably including suppression of NRG1-induced HER2-HER3 dimerization by Tz+Pz (Figure 4b), resulting in HER2-HER4 dimers having a slightly more dominant presence in the overall dimer distribution (Figure 4c). We performed the NRG1 and mAb treatments on a second occasion to verify the reproducibility of the PLA results and our conclusions (Appendix A).

## 4. Discussion

In this study, we used an optimised proximity ligation assay and analysis pipeline to quantify the levels of therapeutically targetable HER2-HER3 dimers in human BM, and the effect of HER2-targeted mAbs on HER2/3 heterodimer complexes in vitro. There were several key findings. As expected, HER2-HER3 dimers were generally more frequent in BM classified as HER2+, but this was not a particularly reliable indicator, as some brain metastases had a high frequency of HER2-HER3 complexes regardless of HER2 status, and conversely, some tumours classified as HER2+ had relatively lower levels. This may be considered a recent finding that HER2-low breast cancers metastasise to the brain at a frequency between that of HER2-negative and -positive tumours [60].

Our analysis of a second, clinically annotated breast cancer BM cohort showed that BM had a higher overall frequency of HER2-HER3 dimers compared to brain-metastatic primary tumours (particularly HER2+ cases, but not exclusively). Interestingly, a subgroup analysis of patient-matched, successive BM showed that HER2-HER3 complexes were significantly less frequent in subsequent recurrences. We do not know the clonal relationships between the first and second BM, but if the latter arose by BM self-seeding, this could suggest there is less selection pressure to maintain HER2-HER3 signalling after initially colonising the brain. 

BM from patients previously treated with HER2-targeted therapy had higher HER2-HER3 dimer levels than treatment-naïve tumours. This trend was modest in our exploratory case series (*p* = 0.1), but given that most HER2+ breast cancer patients are now treated in the first-line setting with trastuzumab and pertuzumab, and that intracranial disease is relatively insensitive to HER2 therapy [12,13,14,15,16], larger cohort studies would be warranted to investigate HER2-HER3 dimerization in relation to acquired resistance, and whether this can be suppressed by pertuzumab and/or anti-HER3 mAbs in the metastatic setting. The NRG-HER3-HER2 axis has been strongly implicated in HER2+ breast cancer relapse [10,17,27,30,32,61], and preclinical studies have shown that pertuzumab and LJM716 (HER3-inactivating mAb) significantly reduce the outgrowth of experimental HER2+BM [27]. Further clinical development of LJM716 is underway [62,63], though it is unclear if planned clinical trials will assess intracranial progression as a specific endpoint.

Our in vitro experiments suggested that in the context of neuregulin oversupply, trastuzumab and pertuzumab are not particularly effective at reducing HER2-HER3 dimerization. HER2 dimerization blockade is pertuzumab’s primary mode of action, and yet the association between neuregulin-1 and the HER2-HER3 complex is about two orders of magnitude stronger than that between pertuzumab and HER2 (equilibrium dissociation constants (K_D_) 0.02 vs. ~2.5 nM, respectively [33,64]). Schwarz and colleagues found that induction of neuregulin-1 and autocrine activation of HER3-PI3K mediates acquired resistance to T-DM1 in HER2+ breast cancer xenografts [61]. In our experiments, HER2-HER3 and HER2-HER4 remained the predominant heterodimer complexes in the presence of NRG1, Tz and Pz. We previously reported frequent co-expression of pHER3 and pHER4 in BM [30,31]. Together with the well-known functional redundancy in the HER family, and frequent alterations in *PTEN* and *PI3K* in BM [30,52,53,65], these observations may help to explain their efficient acquisition of resistance to existing targeted agents.

PLA likely captures receptor clusters within organised membrane signalling patches as opposed to individual dimer complexes [66,67], therefore should be regarded as semi-quantitative. Nevertheless, for relative quantification, it is still a biologically and functionally relevant readout of dimer frequency. This study was not without limitations, and independent analyses would help to corroborate our conclusions; particularly studies that characterise dimer heterogeneity and establish whether particular HER2 or HER3 complexes predict pertuzumab efficacy in animal models or clinical trial biospecimens. 

It would be of great interest to include HER3 homodimers in such studies, which we were unable to analyse because the PLA protocol is not suitable for homodimer analysis. This is because PLA relies on two different species-specific primary antibodies that are at proximity to detect protein–protein interactions. Unless the pair of antibodies are specific against the same epitope, it is not possible to exclude that the PLA signals from two antibodies against the same protein reflect overall protein expression. In vitro single particle tracking studies indicated that HER3 homodimers are more stable than HER2-HER3, and that multiple signalling-competent HER3 homodimers could be catalysed by a single HER2-mediated transphosphorylation event, making HER3-PI3K activity less reliant on equivalent levels of other dimerization partners [66]. If found to be prevalent in BM, this would have important implications for therapeutic development and clinical trial design.

## 5. Conclusions

Coupled with automated digital image analysis, the PLA workflow described here is a straightforward, accurate technique for relative quantification of HER family heterodimers in both cell lines and FFPE tissue samples. Our application of this pipeline to annotated clinical samples showed that HER2-HER3 complexes are prevalent and induced in BM, but not reliably related to HER2 amplification, expression or activation, raising questions about whether these biomarkers would be relevant predictive indicators in BM. To resolve this, preclinical HER-targeted drug efficacy studies may benefit from including dimer quantification as a covariate. 

Importantly, we found that trastuzumab and pertuzumab had minimal impact on the HER2/3 heterodimer landscape in the context of neuregulin oversupply; a unique feature of brain tissue that we simulated by supplementing HER2+ cell cultures with recombinant protein. Our results suggest that neuregulin-1 could compete with HER2 dimerisation blockers by virtue of its high affinity for the HER2-HER3 complex. Hence, in the setting of brain-metastatic disease, antibody–drug or –radiotherapeutic conjugates that target nonspecific payloads to these receptors, rather than relying on signalling blockade, may be favourable to conventional agents. More preclinical studies in this area are warranted.

## Figures and Tables

**Figure 1 cancers-14-00533-f001:**
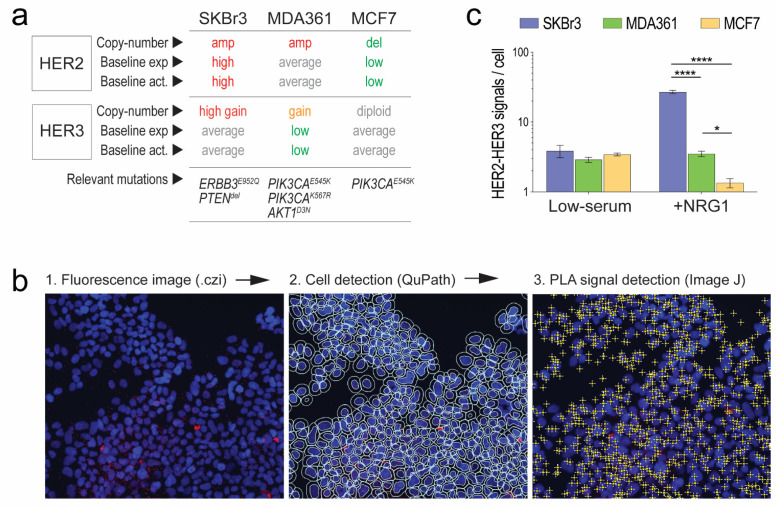
Validation of fluorescent in situ PLA and data extraction pipeline. (**a**) Gene copy number, baseline exp(ression) and act(ivation) status and relevant pathway mutations for the cell lines used. (**b**) Summary of data extraction steps (see methods and Appendix A). (**c**) Comparative analysis of HER2-HER3 complex signals across cell lines in low-serum or NRG1-supplemented media (**** *p* < 0.0001; * *p* < 0.05; Fisher’s LSD test). 20× magnification.

**Figure 2 cancers-14-00533-f002:**
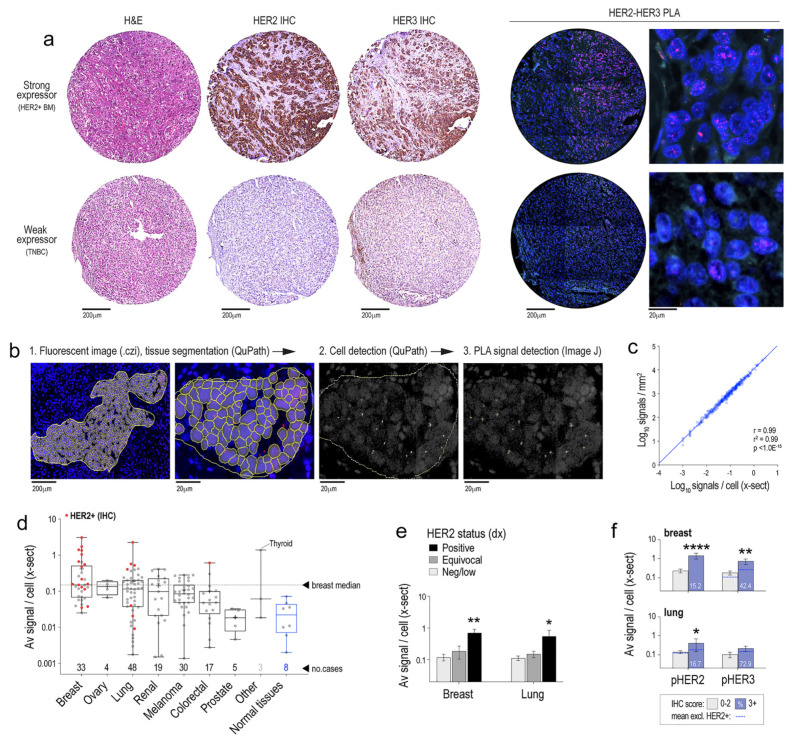
Quantification of HER2-HER3 dimers in a mixed cohort of human brain metastases. (**a**) Representative images from matching TMA cores stained with haematoxylin and eosin (H&E), HER2 or HER3 IHC protocols, or the HER2-HER3 PLA protocol. (**b**) Summary of data extraction steps (see methods and Appendix A). (**c**) Correlation between PLA signals as a function of tissue area or cell number (Spearman correlation (r), goodness of fit (r^2^) and p-value indicated). (**d**) Comparative analysis of HER2-HER3 signal frequency in BM according to primary histology. Cases classified as HER2+ using diagnostic IHC criteria are shown in red. (**e**) Comparison of HER2-HER3 signal frequency in breast and lung cancer BM classified HER2+, equivocal (2+) or negative-low. (**f**) Relationships between HER2-HER3 signal frequency in breast and lung cancer BM with strong activation of HER2 or HER3 (pHER2/pHER3 3+ by IHC) versus negative to moderate activation (0-2+). Dotted blue lines: average signal frequency excluding HER2+ tumours. Fisher’s LSD tests were used for e-f (reference: neg/low groups; * *p* < 0.05, ** *p* < 0.01, **** *p* < 0.0001).

**Figure 3 cancers-14-00533-f003:**
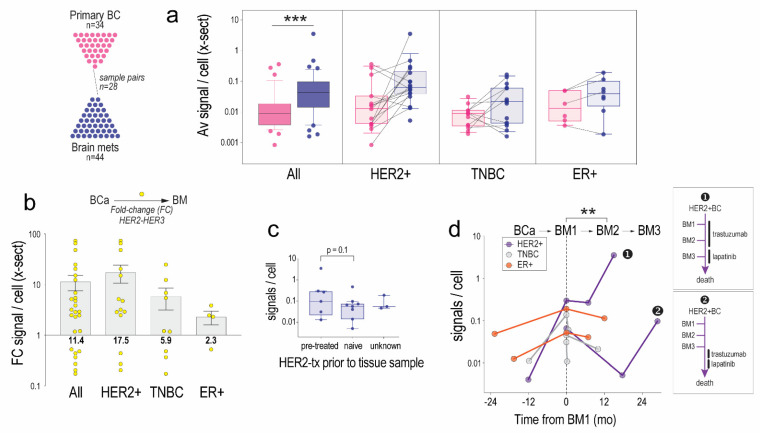
Analysis of HER2-HER3 dimers in brain metastatic breast tumours and breast cancer BM. (**a**) Average HER2-HER3 signals per cell for the entire cohort (all), and disease subtypes (Mann–Whitney, *** *p* = 2.1 × 10^−4^). (**b**) Fold-change (FC) dimer signals between the primary and BM samples of 28 matched pairs (mean ± standard error shown. (**c**) Relationship to HER2-targeted treatment (tx). Marginally higher HER2-HER3 dimer frequency in pre-treated vs. treatment-naïve HER2+BM (*p* = 0.1; 2-tailed *t*-test). (**d**) HER2-HER3 dimer frequency differences between first (BM1) and subsequent (BM2, BM3) brain recurrences for seven patients who had more than one neurosurgical resection procedure. Insets show the timing of HER2-targeted therapy with respect to BM1/2/3 samples. HER2-HER3 dimer frequency in BM2 was lower than the BM1 group (paired, 2-tailed test, ** *p* < 0.01, *** *p* < 0.001).

**Figure 4 cancers-14-00533-f004:**
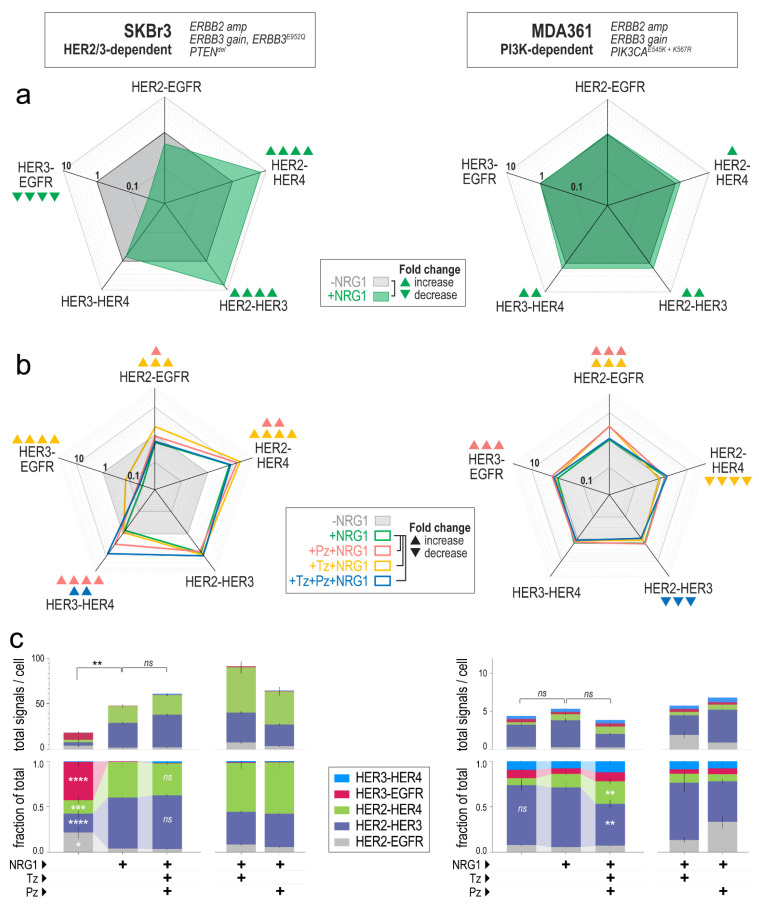
Effects of neuregulin-1, trastuzumab and/or pertuzumab on the HER2/HER3 heterodimer landscape. (**a**) NRG1-induced dimerization in SKBr3 or MDA361 cells (radar plots show fold change). Arrows indicate the direction of change relative to untreated cells (▲ increase, ▼ decrease), where the change was at least 1.5-fold with *p* < 0.05 by 2-tailed *t*-test (▲▲▲▲, *p* < 0.0001, ▲▲▲, *p* < 0.001; ▲▲, *p* < 0.01; ▲, *p* < 0.05). (**b**) NRG1-induced fold change in dimer frequency in the presence of trastuzumab (Tz) and/or pertuzumab (Pz). Arrows indicate the direction of change relative to the +NRG1 reference, where the change was at least 1.5-fold with *p* < 0.05 (2-tailed *t*-test, arrow markers as for (**a**)). (**c**) Changes in the overall distribution of the five complexes in the presence of NRG1, Tz and Pz. Data shown are means +/− standard error, expressed as either absolute (top) or fractional (bottom) dimer signals. Significant changes are highlighted, where changes were ±10% of the reference (+NRG1), with *p* < 0.05 by 2-tailed *t*-test: **** *p* < 0.0001, ***, *p* < 0.001; **, *p* < 0.01; *, *p* < 0.05; *ns*, not significant). Changes with individual mAbs are shown for comparison.

**Table 1 cancers-14-00533-t001:** Primary antibodies, pairing, and optimised conditions.

Dimer Complex	Antibody 1 (Host: Rabbit)				Antibody 2 (Host: Mouse)				Conditions *
	Supplier (Cat No.)	Clone	Host	Dilution	Supplier (Cat No.)	Clone	Host	Dilution	
**In vitro** **Experiments (Fixed Cells on Coverslips)**									
(1)HER3-(2)EGFR	CST (12708)	D22C5	Rabbit	1/100	SC (sc73511)	R2	Mouse	1/500	O/N, 4 °C
(1)HER3-(2)HER2	CST (12708)	D22C5	Rabbit	1/100	Abcam (ab16901)	3B5	Mouse	1/2000	O/N, 4 °C
(1)HER3-(2)HER4	CST (12708)	D22C5	Rabbit	1/100	SC (sc-71070)	3F168	Mouse	1/100	O/N, 4 °C
(1)HER2-(2)EGFR	DAKO (A0485)	Poly	Rabbit	1/2500	SC (sc73511)	R2	Mouse	1/500	O/N, 4 °C
(1)HER2-(2)HER4	DAKO (A0485)	Poly	Rabbit	1/2500	SC (sc-71070)	3F168	Mouse	1/100	O/N, 4 °C
**In situ Analysis (FFPE Tissue Sections)**									
(1)HER2-(2)HER3	DAKO (A0485)	Poly	Rabbit	1/2500	MP (05-390)	2F12	Mouse	1/200	10 mM Tris-EDTA (pH 8.8) 95 °C; 30′ O/N 4 °C

* Abbreviations: CST, Cell Signaling Technology; O/N, overnight; Poly, polyclonal; MP, Millipore; SC, Santa Cruz.

## Data Availability

The data presented in this study are available in this article (and Appendix A).

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
