# Peer review of "Landscape of Epidermal Growth Factor Receptor Heterodimers in Brain Metastases"

_cancers, 2022, doi:10.3390/cancers14030533_

Round 1
Reviewer 1 Report
I congratulate the authors for this work. Really interesting, well written, well presented and definitely worth a release in Cancers.
I have no notes to make either from a scientific point of view or on the presentation of the study, which is already well articulated.
My only suggestion is on the quality of the images, which does not do justice to the quality of the work, I recommend increasing the resolution (at least 600 dpi), in order to make the work even more attractive to the reader.
Author Response
Thank you for reviewing our manuscript, please see our responses in the attachment.

Reviewer 2 Report
The authors address the important clinical issue of HER3 expression associated with drug resistance in solid cancers. They approached this challenge by asking whether HER2-HER3 heterodimerization underlies this resistance, specifically in brain metastases from breast cancer. The authors first validated proximity ligation assay (PLA) protocol for HER2-HER3 heterodimers using breast cancer cells lines with high (SKBr3), average (MDA361), and low (MCF7) ERBB2/ERBB3 expression. PLA signal per cell was detected in ImageJ following cell detection in QuPath and was seen correlate with ERBB2/ERBB3 expression in these cells following NRG1 stimulation. The authors then took FFPE TMA sets with brain mets from mixed cancers and another set with breast cancer (of which, there were 28 pairs of primary-brain met samples). Following initial annotation of tumor rich regions, the authors performed PLA for HER2-HER3 dimers, and developed an automated pipeline with QuPath and ImageJ. The authors then compared the HER2-HER3 dimerization frequency for primary breast cancer and the corresponding brain mets. Lastly, they performed in-vitro experiments to understand the effects of neuregulin1 stimulation and transtuzumab/pertuzumab on the HERx heterodimer landscape.
The authors present the results in a well written manuscript. The experiments are designed well, and data presented succinctly. The main findings of this paper is that there is relatively higher HER2-HER3 dimerization in brain mets with some correlations with HER2 expression or activation, and HER3 activation. In HER2-HER3 dependent SKBr3 cell lines, neuregulin 1 stimulation resulted in increased HER2-HER3 dimerization that was relatively unchanged with trastuzumab/pertuzumab. In PI3K dependent MDA361 cell line, slight increase in HER2-HER3 dimerization was observed with stimulation. These data suggest that neuregulin may drive HER2-HER3 dimerization in brain mets that is not responsive to trastuzumab, pertuzimab or combinatorial therapy.
This paper extends the work initially presented in Saunus et al. (2015). Data including HER2 and HER3 activation status in brain mets was used from the prior analysis. The addition of PLA to detect HER2-HER3 dimers and creating a pipeline for such analysis using TMA are the primary contributions of this paper.
I have two major concerns:
- Predominantly nuclear staining with anti-HER3 (2F12) antibody used for PLA studies for FFPE tissues. In Figure S1, sample images from SKBr3 cells are quite distinct for the tow antibodies shown (D22C5 vs. 2F12). The authors need to demonstrate the validity of using 2F12 antibodies to convince readers about the accuracy of the FFPE PLA data.
- Baseline HER3 expression. In Figure 2, the authors demonstrate representative TMA samples with HER3 staining. Have the authors performed a larger survey of HER3 expression in TMA samples? Is HER2-HER3 dimerization correlated with and dependent in HER3 expression (rather than HER3 activation). I recommend that the authors perform and quantify HER3 expression in primary vs. brain met samples to clarify this potential association.
I have two minor concerns:
- Following the ‘Conclusion’ section, there are additional, superfluous discussion (lines 417-420).
- Analysis of fold change data presented in Figure 3b. Please explain the analysis strategy used to calculate and present the data shown. How is the mean calculated and is there any statistically significant difference between groups?
Author Response

(The authors gave the same response as above.)

Reviewer 3 Report
Malcom Lim et al proposed an original articled entitled : Landscape of Epidermal Growth Factor Receptor Heterodimers in Brain Metastases.
In general, the articles is well introduced and referenced. The main goal was to optimize a PLA (proximity ligation assay) pipeline for analysis of HER2-HER3 complexes. Authors thus used this PLA pipeline to assess variation between primary tumors and brain metastasis (BM) and their correlation with anti-HER2 resistance, commonly reflected by BM.
I suggest only one major limitation : the title and the whole MS are focused on the interrelation HER dimers, differences between primary tumors and their matched BM. However the conclusion do not summarize the MS and only focused on the technique. The clinical application should be mentioned. Please revise it.
I also suggest some minor modifications such as :
Table 1 should be moved in additional section
Any antibody drug conjugates are discussed. I recommend to introduce some discussion (limitation paragraph i.e) : TDM1 and TDxD could have different biological results on these experiments.
Thanks to the editorial team and the authors for this review request.
Author Response

(The authors gave the same response as above.)
